# Potent kinase inhibitors from the Merck KGaA OGHL: Novel hits against *Trypanosoma brucei* with potential for repurposing

Darline Dize[1]*, Aurélien F. A. Moumbock[2], Vianey C. Tchuenguia[1], Germaine Y. Bougnogolo[1], Fride S. B. Nana[1], Sandra D. W. Monkam[1], Fabrice F. Boyom [1,3]*

**1** Antimicrobial and Biocontrol Agents Unit (AmBcAU), Department of Biochemistry, Faculty of Science, University of Yaoundé 1, Yaoundé, Cameroon, **2** Faculty of Chemistry and Pharmacy, Institute of Pharmaceutical Sciences, University of Freiburg, Freiburg, Germany, **3** Department of Drug Discovery, Advanced Research and Health Innovation Hub (ARHIH), Yaoundé, Cameroon

* fabrice.boyom@fulbrightmail.org (FFB) darline.dize@yahoo.fr (DD)

## Abstract

African trypanosomiasis remains a critical public health concern, with over 55 million people still at risk of infection. There are several issues associated with the current therapies including toxicity and resistance, which represent the main bottleneck of trypanosomiasis control. Thus, it is urgent to develop novel therapeutic tools with distinct mechanisms of action. The *in vitro* phenotypic screening of the Merck KGaA Darmstadt German Open Global Health Library (OGHL) against *Trypanosoma brucei brucei* yielded three potent kinase inhibitors belonging to different chemical series: a phenylcarbonylacrylamide (OGHL00006); a 2,4-di(phenylamino)pyrimidine (OGHL00133); and a 3-(triazol-4-yl)-7-azaindole (OGHL00169). They exhibited low micromolar to nanomolar median inhibitory concentrations (IC$_{50}$ values of 0.6 μM, 0.007 μM, and 0.25 μM, respectively) and good selectivity when tested on Vero cells (SI > 2). OGHL00006 and OGHL00169 induced a rapid and irreversible growth arrest of *T. b. brucei* within 4–24 hours of incubation. Interestingly, these two hits have also been reported to display antiplasmodial and/or anthelminthic activities, hinting at a similar mechanism of action across multiple species. Given the significant sequence similarities between the human and trypanosome kinomes, we rationalized the putative mechanisms of action for the identified hits through comparative modeling of protein–ligand complexes. This study suggests promising avenues for drug and/or target repurposing against trypanosomiasis.

**Data availability statement:** All data are provided in the manuscript.

**Funding:** The author(s) received no specific funding for this work.

**Competing interests:** The authors have declared that no competing interests exist.

## Author summary

African trypanosomiasis is a parasitic disease caused by parasites of the *Trypanosoma brucei* species, affecting both human and animal health thereby hindering socio-economic development in endemic countries. Current anti-trypanosomal therapies are compromised by issues of drug resistance and toxicity. Moreover, in contrast to the progress made in the treatment of human trypanosomiasis, no new drug has been developed for animal trypanosomiasis in several decades, highlighting an urgent need for novel therapeutic options. Given the high cost, length and complexity of conventional drug development, repurposing existing compounds has become a widely adopted strategy in drug discovery. In this study, we performed an antitrypanosomal screening of the Merck KGaA Darmstadt Germany Open Global Health Library and identified three promising and non-toxic compounds: OGHL00006, OGHL00133 and OGHL00169. Subsequent investigations demonstrated that only OGHL00006 and OGHL00169 exhibit trypanocidal activity. Molecular modelling further revealed that these two compounds inhibit key Trypanosomes kinases involved in parasite growth and survival. This study uncovers two potential lead compounds offering promising candidates for the development of new therapies against African trypanosomiasis.

## Introduction

Human African trypanosomiasis (HAT), commonly known as sleeping sickness, is a neglected tropical disease caused by *Trypanosoma brucei* parasites and transmitted by tsetse flies. Despite advances in disease control through vector management and public health interventions, HAT remains a significant health challenge in sub-Saharan Africa, especially in rural communities with limited access to healthcare [1]. Similarly, animal African trypanosomiasis (AAT), known as nagana, devastates livestock populations, imposing severe economic constraints on agriculture-dependent communities [2]. HAT and AAT treatments have long been constrained by a limited arsenal of therapeutic options, many of which have been in use for decades. For instance, HAT regimens historically relied on melarsoprol, an arsenical compound with severe adverse reactions including fatal encephalopathy in a significant percentage of cases (5–10%) [3]. Recent advancements, such as the introduction of fexinidazole, a convenient oral therapy for early and late-stage gambiense-HAT, have provided safer alternatives [4,5]. Acoziborole, an oral oxaborole compound has emerged in 2021 a promising single-dose therapy for HAT caused by *T. b. gambiense.* With demonstrated efficacy in both early and late stages of the disease and an excellent safety profile as depicted by a phase 2/3 trial, acoziborole represents another significant milestone in HAT treatment [6]. However, both compounds efficacy is limited to *T. b. gambiense*, leaving untreated infections caused by the more virulent *T. b. rhodesiense*. In contrast to HAT, AAT treatments remain reliant on

decades-old compounds, such as diminazene aceturate and isometamidium chloride, which are associated with resistance issues and narrow therapeutic window, emphasizing the urgent need for innovative approaches to address both human and animal trypanosomiasis. Unfortunately, none of the newly developed compounds have been demonstrated to address AAT [7]. Thus, the lack of broad-spectrum trypanocidal drugs and the emergence of drug resistance in both human and animal trypanosomiasis highlights the pressing need for novel therapeutic agents. Phenotypic screening has re-emerged as an effective tool for discovering drugs against neglected tropical diseases such trypanosomiasis. Unlike target-based approaches, which focus on specific molecular pathways, phenotypic screening evaluates compounds in the context of whole parasites, capturing diverse biological effects (on-target and off-target) and revealing new mechanisms of action [8,9]. Complementary to this, drug and/or target repurposing has garnered attention as a way to accelerate drug discovery by identifying compounds with potential activity against parasites that were initially developed for other diseases [10,11]. For instance, some antagonists to human kinases or G-protein-coupled receptors (GPCRs) have shown promise in disrupting essential pathways or proteins found in *Leishmania donovani*, *T. b. brucei*, and *T. cruzi*, making them attractive candidates for repurposing [12]. Therefore, this strategy holds the potential to identify starting point for the discovery of alternative medicines. With this perspective, a growing number of pharmaceutical industries and some non-profit organizations provide access to libraries of well characterized compounds to researchers to address some unmet medical needs including HAT. In the present study, the Merck KGaA Open Global Health Library (OGHL), a collection of 250 bioactive molecules targeting various pathways, such as hormone or neurotransmitter (serotonin, angiotensin II, endothelin, etc.) receptors, ion channels, enzymes (phosphodiesterases, proteases, and phosphatases) and several others kinases was screened to identify compounds with potent antitrypanosomal activity [13,14] based on the hypothesis that repurposing the OGHL and its kinase inhibitors may hold potential as a valuable approach to unveil novel antiparasitic scaffolds with appropriate safety and pharmacokinetic properties.

## Methods

### Compound library

Merck KGaA, Darmstadt, Germany, has established an Open Innovation initiative to support researchers from universities, research institutes, and private companies for research and development purposes in the field of infectious diseases. As part of this initiative, they provide the Open Global Health Library (OGHL) free of charge, a collection of compounds that have reached various stages of development, including clinical Phase II. For this study, the OGHL was made available upon request through Merck's Open Innovation Portal (Healthcare Mini Library) [13,14]. The compounds were supplied as 10 mM DMSO stock solutions (30 μL) in 384-well microplates.

### Potency assessment of compounds against *T. b. brucei*

Bloodstream forms of *T. b. brucei* Lister 427, a virulent laboratory strain of *Trypanosoma brucei* were cultured in HMI-9 medium as previously described [15]. The antitrypanosomal activity of the compounds was evaluated using the resazurin reduction assay which relies on the conversion of resazurin (blue, non-fluorescent) to resorufin (pink, fluorescent) by mitochondrial dehydrogenases in metabolically active parasites. The resulting signal measured between 570 nm and 590 nm, is directly proportional to the quantity of viable parasites [16]. Prior to the actual determination of median inhibitory concentrations ($IC_{50}$s), parasites (90 μL) at a density of $2 \times 10^5$ cells/mL were incubated in duplicate with 10 μL of each compound at a single concentration of 10 μM in 384-well plates for 72 hours. Pentamidine isethionate (1 μM) served as the reference drug (positive control), while untreated parasites and parasites exposed to 0.1% DMSO were included in assay plates as negative and vehicle controls respectively. After incubation, a working solution of resazurin (0.15 mg/mL in DPBS) was added to the plates in a 1:10 (v/v) ratio, followed by incubation for 4 hours at 37 °C with 5% $CO_2$. Fluorescence was measured using an Infinite M200 microplate reader (Tecan) at excitation and emission wavelengths of 530 nm

and 590 nm, respectively [17]. Compound achieving ≥ 50% inhibition of trypanosomes were selected for further dose-response studies.

## Antitrypanosomal dose-response studies of the compounds

Compounds that adhered to the cutoff criterion (≥ 50% inhibition) were serially diluted and tested for their half-maximal inhibitory concentrations ($IC_{50}$). The concentrations range tested was 10–0.000128 µM for the compounds and 1–0.0000128 µM for the reference drug pentamidine isethionate. Dose-response curves were generated using GraphPad Prism software (version 8.0).

## Mammalian cell culture and cytotoxicity assays

Vero E6 (African green monkey kidney), Raw264.7 (murine macrophages) and HepG2 (human hepatocyte carcinoma) cell lines, sourced from the American Type Culture Collection, were cultured in DMEM supplemented with 10% (v/v) heat-inactivated fetal bovine serum, 1% (v/v) non-essential amino acids, and 1% (v/v) penicillin-streptomycin. Cells were maintained at 37 °C in a 5% $CO_2$ atmosphere. For cytotoxicity testing, they were seeded (90 µL) in 96-well plates at a density of 1x10$^4$ cells/well, in duplicate, and allowed to adhere overnight. Serial dilutions (50-0.08 µM) of the compounds were prepared, and 10 µL of each were incubated with the monolayer for 48 hours. Podophyllotoxin was tested as a positive control at 10 µM and 0.5% DMSO (100% cell viability) was used as negative control. Cell viability was then assessed using the resazurin reduction assay, as described above. The 50% cytotoxic concentration ($CC_{50}$) was calculated using GraphPad Prism software (version 8.0), and selectivity indices (SIs) were determined by dividing the *T. b. brucei* $IC_{50}$ by the $CC_{50}$.

## Confirmatory dose-response studies

Three compounds were selected for confirmatory studies based on their highest antitrypanosomal activity ($IC_{50}$ ≤ 1µM) and good selectivity (SI ≥ 30). These compounds were re-supplied as powdered forms by Merck KGaA, Darmstadt Germany, dissolved in DMSO 100%, and tested under identical experimental conditions. For the antitrypanosomal assays, concentrations ranged from 10–0.000128 µM for the test compounds and 1–0.0000128 µM for the reference drug pentamidine isethionate. For the cytotoxicity experiments, concentrations ranged from 50–0.08 µM for the test compounds and 10–0.00064 µM for the reference compound Podophyllotoxin.

## Trypanosome killing kinetics

To evaluate the killing kinetics of selected compounds, bloodstream forms of *T. b. brucei* (2 x 10$^5$ cells/mL) were incubated in normal culture conditions in the presence of compounds at 8$IC_{50}$, 4$IC_{50}$, $IC_{50}$, and 0.25$IC_{50}$ concentrations in 24-well plates. Upon incubation, cultures were centrifuged at 0, 4, 8, 12, 24, 30, 36, 48, 60, and 72 hours' time intervals at 2,500 rpm followed by motile trypanosomes enumeration using a Lumascope LS520 inverted microscope [17]. To determine the mode of inhibition (parasitostatic or parasiticidal), a rescue experiment was conducted. Trypanosomes were exposed to the test doses (8$IC_{50}$, 4$IC_{50}$, $IC_{50}$, and 0.25$IC_{50}$) of the compounds for 4 hours. They were then washed three times with phosphate buffer saline (DPBS 1X), centrifuged at 2,500 rpm for 7 minutes, and resuspended in complete drug-free medium. These parasites were cultured under the same conditions, and motile organisms were enumerated with the Neubauer chamber at regular intervals over 72 hours. Data were plotted using GraphPad Prism 8.0 to determine the inhibition kinetics and classify the inhibitors as cidal or static. Pentamidine isethionate served as the reference compound and experiments were performed in duplicate.

## Molecular docking

Crystal structures were retrieved from the RCSB Protein Data Bank (PDB; https://www.rcsb.org/) and modeling studies were performed with the Schrödinger 2024–4 software (Schrödinger LLC, NY, USA) as previously described [18,19]. The

homology models of *Tb*KFR1 (Tb927.10.7780) and *Tb*PDK1 (Tb927.9.4910) with transplanted ligands were built with Prime [20], using as templates the ligand-bound cocrystal structures of their human homologs MAPK1 (PDB ID: 6G54) and PDK1 (PDB ID: 3RCJ), respectively. The crystal structure of *Tb*CLK1 (Tb927.11.12410; PDB ID: 6Q2A), and homology models of *Tb*KFR1 and *Tb*PDK1 were prepared with the Protein PrepWizard [21], meanwhile the structures of the identified hits (OGHL00006, OGHL00133, and OGHL00169) were prepared with LigPrep (Schrödinger LLC, NY, USA), using the OPLS4 force field [22]. The geometric center of the cocrystallized or transplanted ligand was considered as the docking grid centroid; and Cys215 of *Tb*CLK1 was selected as the reactive residue for the SMARTS-based Michael addition reaction. Core-constrained noncovalent docking was carried out using Glide [23] while position-constrained covalent docking was carried out using CovDock [24]; both with the SP (Standard Precision) scoring function.

### Molecular dynamics simulations

MD (molecular dynamics) simulations were carried out using the program Desmond [25] with the OPLS4 force field [22]. Each noncovalent protein–ligand docked complex was solvated with TIP3P water molecules while neutralizing the charge of the system with counter ions, within an orthorhombic periodic box with 10 Å side barriers. After an equilibration protocol, the production of MD simulations was conducted for 100 ns in an NPT ensemble at 300 K regulated by a Nosé–Hoover thermostat and a Martyna–Tobias–Klein barostat. Atomic coordinates were recorded at an interval of 400 ps, for a total 250 frames.

### Statistical analysis

Data were analysed using Microsoft Excel and GraphPad Prism 8.0. Nonlinear regression with a sigmoidal dose-response fit model was applied to the data to determine $IC_{50}$ and $CC_{50}$ values. Results represent the mean ± standard deviation (SD) from two independent experiments performed in duplicate.

## Results

### Compounds from the OGHL are efficacious against the bloodstream forms of *T. b. brucei in vitro*

Screening the OGHL compounds at a single concentration of 10 μM against the bloodstream forms of *T. b. brucei* showed that from the 250 compounds, 29 compounds exhibited trypanosome growth inhibition > 50%. The Z' factor was calculated for each screen to ensure the reliability of the data and as previously outlined [26]; a Z' score of 0.85 was an indication of an excellent screening as in the present study (Fig 1).

The 29 compounds were taken forward to a 7-point dose-response growth inhibition assay against the bloodstream form of *T. b. brucei* Lister 427 and their cytotoxic effects on HepG2 mammalian cells were evaluated in parallel. The results (Table 1) are summarized in terms of $IC_{50}$ (concentration required to inhibit 50% of parasite growth), $CC_{50}$ (concentration required to induce 50% cytotoxicity in HepG2 cells), and the selectivity index (SI, calculated as $CC_{50}/IC_{50}$). Compounds with an SI > 10 were considered highly selective toward the trypanosomes.

Several compounds exhibited strong antitrypanosomal activity, with $IC_{50}$ values below 1 μM. The compound OGHL00133 emerged as the most potent, with an $IC_{50}$ of 0.0051 μM and the highest selectivity (SI > 9779), compared to an $IC_{50}$ of 0.0019 μM for pentamidine (the reference drug). Other notable compounds include OGHL00169 ($IC_{50}$ = 0.85 μM, SI = 58.82) and OGHL00006 ($IC_{50}$ = 0.56 μM; SI = 31.25). Moderately potent compounds, such as OGHL00134, OGHL00138, OGHL00227, and OGHL00250 showed $IC_{50}$ values between 1 and 3 μM and SI > 10. In contrast, some compounds such as OGHL00208 and OGHL00238 demonstrated low selectivity (SI of 0.31 and 0.75, respectively), indicating they are more toxic to mammalian cells than effective against parasites.

Among the identified hits, compounds OGHL00006, OGHL00133, and OGHL00169 displayed the highest potency ($IC_{50}$ ≤ 1 μM) and selectivity (SI ≥ 30). Subsequently, their potency was confirmed upon retesting using powdered

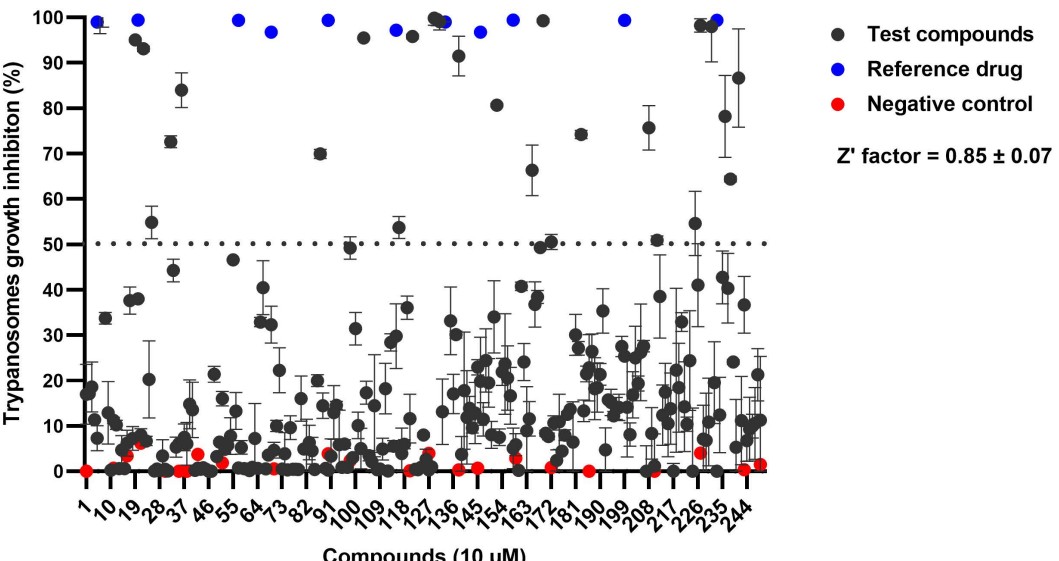

**Fig 1. Inhibition percentage of hits identified in a primary antitrypanosomal screen of the OGHL.** Compounds were tested for activity at 10 µM against bloodstream forms of *T. b. brucei*. Data are presented as mean % inhibition of trypanosomes growth + SD (n = 2). Compounds which inhibited the growth by > 50% (above the lower dotted line) are shown.

compound forms. However, the activity data revealed notable differences, particularly in selectivity between the DMSO stock solutions and the powdered forms of these compounds. This discrepancy may result from differences in compound stability, purity, or solubility between the DMSO stock solutions and powdered forms (Table 2 and Fig 2).

### Structure-activity relationship (SAR) analysis

The 29 antitrypanosomal hits belong to diverse chemical classes and SAR analysis revealed some trends associated with six of them: 5-(tetrahydroquinolin-6-yl)-3,6-dihydrothiadiazinones (OGHL0025 and OGHL0032), 2,3-diphenylthiazo lidinones (OGHL00152 and OGHL00183), bisphenylureas (OGHL00129, OGHL00130, and OGHL00131), 2,4-di (phenylamino)pyrimidine (OGHL00133 and OGHL00134), tetrahydrotriazolopyridines (OGHL00208 and OGHL00211), and imidazopyridine/imidazothiazoles (OGHL00231, OGHL00241, OGHL00238, OGHL00225 and OGHL00227) (Table 1 and Fig 3).

   In the first series (5-(tetrahydroquinolin-6-yl)-3,6-dihydrothiadiazinones), the conversion of a carboxamide in OGHL0025 to an ethylamidine in OGHL0032 resulted in a slight increase in activity and reduced toxicity. In the 2,3-diphenylthiazolidinone series, O-demethylation of OGHL00183 ($IC_{50}$ = 5.04 µM) had no significant impact on activity as exemplified with OGHL00152 ($IC_{50}$ = 8.87 µM). For bisphenylureas, replacing the pendant azole ring in OGHL00130 ($IC_{50}$ = 3 µM, SI = 4.2) with a 1,2,4-triazole led to a 5-fold increase in activity and a 2-fold increase in selectivity for OGHL00129 ($IC_{50}$ = 0.6 µM; SI = 9.5). Moreover, the removal of an *N*-methylcarboxmide moiety led to a decrease in selectivity despite an increase in potency for OGHL00131 ($IC_{50}$ = 0.46 µM; SI = 6.6). In the 2,4-di(phenylamino)pyrimidine series, replacement of a bulky N-morpholino group in OGHL00134 ($IC_{50}$ = 1.3 µM; SI = 16.84) with a methoxy group induced a huge increase in both activity and selectivity for OGHL00133 ($IC_{50}$ = 0.005 µM; SI > 9774).The tetrahydrotriazolopyridine (R/S)-racemate, OGHL00208 ($IC_{50}$ = 7.53 µM; SI = 0.31) is less potent and less selective than the (R)-enantiomer OGHL00211 ($IC_{50}$ = 2.37 µM; SI = 2.5) suggesting the latter is most likely the active enantiomer. In the final series, modifying the heterocyclic core from an imidazo[1,2-a]pyridine (OGHL00231: $IC_{50}$ = 9.06 µM; SI > 5.5) to imidazo[2,1-b]thiazole (OGHL00227: $IC_{50}$ = 2.8

**Table 1. Potency data of the 29 inhibitors against bloodstream forms of *T. b. brucei*.**

| OGHL ID | Chemical class | IC$_{50}$±SD (µM) *T. b. b.* | CC$_{50}$±SD (µM) HepG2 | SI |
|---|---|---|---|---|
| OGHL00006 | Phenylcarbonylacrylamide | **0.56±0.001** | **17.5±0.94** | 31.25 |
| OGHL00007 | 5-Hydroxyindole | 5.3±0.7 | >50 | >9.47 |
| OGHL00019 | 2-Phenyl-6-azabenzimidazole | 2.49±0.1 | 19.845±2.72 | 7.97 |
| OGHL00022 | pteridine | 8.7±0.09 | 18.745±0.13 | 2.1 |
| OGHL00025 | 5-(Tetrahydroquinolin-6-yl)-3,6-dihydrothiadiazinone | 9.22±0.0007 | 33.305±2.83 | >5.42 |
| OGHL00032 | 5-(Tetrahydroquinolin-6-yl)-3,6-dihydrothiadiazinone | 5.04±0.16 | 48.43±0.25 | 9.6 |
| OGHL00036 | Imidazopyridinone | 6.98±0.094 | 14.275±0.98 | 2.04 |
| OGHL00087 | Benzoxazolone | 8.62±0.093 | 27.265±4.5 | 3.16 |
| OGHL00103 | Pyrazolopyrimidinone | 7.8±0.07 | 41.5±3.4 | >6.38 |
| OGHL00121 | Tetrahydrobenzothienopyrimidine | 8.13±1.44 | 33.775±1.02 | >6.15 |
| OGHL00129 | Bisphenylurea | 0.608±0.08 | 5.7985±4.6 | 9.5 |
| OGHL00130 | Bisphenylurea | 3.05±0.14 | 12.88±0.5 | 4.2 |
| OGHL00133 | 2,4-Di(phenylamino)pyrimidine | **0.0051±0.00018** | **>50** | >9779 |
| OGHL00134 | 2,4-Di(phenylamino)pyrimidine | 1.31±0.22 | 22.065±5.06 | 16.84 |
| OGHL00138 | 3-(Oxadiazol-2-yl)-7-azaindole | 1.55±0.2 | 23.05±3.57 | 14.87 |
| OGHL00131 | Bisphenylurea | 0.46±0.03 | 3.053±0.015 | 6.6 |
| OGHL00152 | 2,3-Diphenylthiazolidinone | 8.87±0.7 | ND | ND |
| OGHL00169 | 3-(Triazol-4-yl)-7-azaindole | **0.85±0.02** | **50** | 58.82 |
| OGHL00165 | 5-Biphenyloxadiazole | 8.83±0.09 | >50 | >5.63 |
| OGHL00183 | 2,3-Diphenylthiazolidinone | 9.64±0.05 | 50 | 5.2 |
| OGHL00208 | Tetrahydrotriazolopyridine | 7.53±1.27 | 2.36±0.52 | 0.31 |
| OGHL00211 | Tetrahydrotriazolopyridine | 2.37±0.22 | 5.94±2.55 | 2.5 |
| OGHL00225 | Imidazothiazole | 9.86±0.0007 | >50 | >5.07 |
| OGHL00227 | Imidazothiazole | 2.8±0.08 | >50 | >17.85 |
| OGHL00231 | Imidazopyridine | 9.06±0.13 | >50 | >5.5 |
| OGHL00241 | Imidazothiazole | 2.28±0.04 | 9.259±0.019 | 4.05 |
| OGHL00250 | Furopyridine | 2.15±0.2 | >50 | >23.25 |
| OGHL00238 | Imidazothiazole | 2.67±0.7 | 2.01±0.46 | 0.75 |
| OGHL00236 | Imidazopyridine | 6.19±0.4 | 9.07±0.94 | 1.5 |
| Pentamidine isethionate | | 0.00137±0.0003 | ND | ND |
| Podophyllotoxin | | ND | 0.031±0.009 | ND |

IC$_{50}$ and CC$_{50}$ represent the half-maximum growth inhibition concentration against bloodstream form of *T. b. brucei* and the human hepatocellular carcinoma cells (HepG2). Growth inhibition measurements were generated with two independent biological replicates. Values shown are geometric mean±SD. ND: Not determined.

µM; SI > 17.85) led to a 3.5-fold increase in activity and selectivity. However, replacing the methoxy group in OGHL00227 (SI > 17.85) with a cyano group resulted in a 23-fold increase in toxicity for OGHL00238 (SI = 0.75).

## OGHL00006 and OGHL00169 exhibit rapid trypanocidal activities

Time-kill assays were conducted for each compound to evaluate their rate of action against *Trypanosoma b. b.* Fig 4A–C depict the killing curves of OGHL00006, OGHL00133, and OGHL00169, respectively. The results revealed distinct activity profiles for the three inhibitors. OGHL00006 and OGHL00169 displayed strong trypanocidal activity, achieving complete parasite clearance at concentrations ≥ IC$_{50}$, with no regrowth observed over 72 hours at higher concentrations (4IC$_{50}$ and 8IC$_{50}$) (Fig 4A). Notably, OGHL00006 showed a rapid onset of action, with parasite killing observed within 4 hours

**Table 2. Antitrypanosomal activity and selectivity of the DMSO stock solution and powdered forms of the selected inhibitors.**

| Compounds | Primary screen (DMSO stock solution) | | | Confirmation screen (powdered form) | | | | |
|---|---|---|---|---|---|---|---|---|
| | $IC_{50} \pm SD$ (µM) T. b. b. | $CC_{50} \pm SD$ (µM) HepG2 | SI HepG2 | $IC_{50} \pm SD$ (µM) T. b. b. | $CC_{50} \pm SD$ (µM) Raw | $CC_{50} \pm SD$ (µM) Vero | SI Raw | SI Vero |
| OGHL00006 | 0.56±0.001 | 17.5±0.94 | 31.25 | 0.60±0.1 | 2.20±0.05 | 1.20±0.07 | 3.7 | 2 |
| OGHL00133 | 0.0051±0.00018 | > 50 | >9803.9 | 0.007±0.00073 | 4.50±0.10 | 1.01±0.09 | 642.86 | 144.28 |
| OGHL00169 | 0.85±0.02 | 50 | 58.82 | 0.25±0.04 | 12.07±1.60 | 2.80±0.03 | 48.28 | 11.2 |
| Pentamidine isethionate | 0.00137±0.0003 | ND | ND | 0.00185±0.0003 | ND | ND | ND | ND |
| Podophyllotoxin | ND | 0.031±0.009 | ND | ND | 0.016±0.002 | 0.08±0.005 | ND | ND |

$IC_{50}$ and $CC_{50}$ represent the half-maximum growth inhibition concentration against bloodstream form of *T. b. brucei* and the cell lines Raw264.7 and Vero. Growth inhibition measurements were generated with two independent biological replicates. Values shown are geometric mean±SD. ND: Not determined.

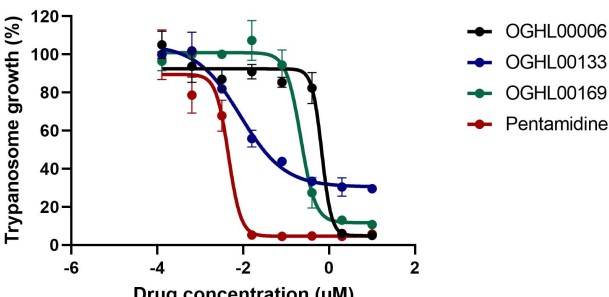

**Fig 2. Dose response curves of selected compounds.** The curves and the $IC_{50}$ values were calculated from readings measured in duplicate and expressed as the % inhibition of trypanosomes growth at each concentration. Deduced $IC_{50}$ values for each test compound are indicated in the Table 2 above.

(Fig 4A), while OGHL00169 required 24–36 hours of exposure to achieve similar effects (Fig 4C). However, at $IC_{50}$ and sub-inhibitory concentrations ($0.25IC_{50}$), partial parasite regrowth was observed, suggesting that prolonged exposure or higher doses may be necessary for complete eradication (Fig 4C). These findings were further corroborated by post-drug exposure monitoring, where parasites were subjected to the compounds for a short duration. The results confirmed the irreversible trypanocidal effects of OGHL00006 and OGHL00169 at their respective $4xIC_{50}$ and $8xIC_{50}$ concentrations, underscoring their potential as cytocidal agents with a rapid onset of action (Fig 5). Considering the goal of selecting the lowest effective concentration that is both selective (minimally toxic) and produces the desired effect, the $4IC_{50}$ concentration appears to be the most appropriate for both compounds. In contrast, OGHL00133 exhibited cytostatic activity, as parasite growth was not completely inhibited at any tested concentration, and parasite density increased over time (Fig 4B). This suggests a delayed death effect without clear trypanocidal activity. Pentamidine, included as a reference drug, demonstrated complete and sustained parasite clearance across all tested concentrations. Its trypanocidal effect was evident within the first 24 hours, reaffirming its strong and rapid efficacy (Figs 4D and 5C).

## Molecular modeling

All three identified *T. b. brucei* inhibitors are known anticancer compounds, specifically kinase inhibitors. Given the significant sequence similarities between the human and trypanosome kinomes [27–30], we rationalized the putative mechanisms of action of these active compounds through comparative modeling of protein–ligand complexes.

**Fig 3. Structures of the diverse compound series emerging from the screening.**

OGHL00133 {2-((5-bromo-2-((4,5-dimethoxy-2-methylphenyl)amino)pyrimidin-4-yl)amino)-N-methylbenzenesulfonamide} is an inhibitor of the focal adhesion kinase (FAK; $IC_{50}=2.8$ nM), anaplastic lymphoma kinase (ALK; $IC_{50}=0.03$ μM), insulin-like growth factor 1 (IGF-1R; $IC_{50}<0.08$ μM), leucine-rich repeat kinase 2 (LRRK2; $IC_{50}=0.13$ nM), zeta-chain-associated protein kinase 70 (ZAP-70), the germinal center-like kinase (GLK, also known as MAP4K3) and the proline-rich tyrosine kinase 2 (PYK2) [31]. Since there are no tyrosine kinases in the trypanosome kinome [27], the putative target of OGHL00133 could most likely be a parasitic homolog of the serine/threonine kinase MAP4K3. Interestingly, the conditional knockdown of *Tb*KFR1 (*Tb*MAPK1; Tb927.10.7780) in bloodstream form by RNA interference (RNAi) leads to slow parasitic growth [32], the same phenotype observed upon *T. b. brucei* inhibition by OGHL00133. Hence, a homology model of *Tb*KFR1 was built based on the cocrystal structure (PDB ID: 6G54) of *h*MAPK1 (*h*ERK2) in complex with SM1–71, a ligand sharing a 5-halo-2,4-di(phenylamino)pyrimidine scaffold with OGHL00133. Next,

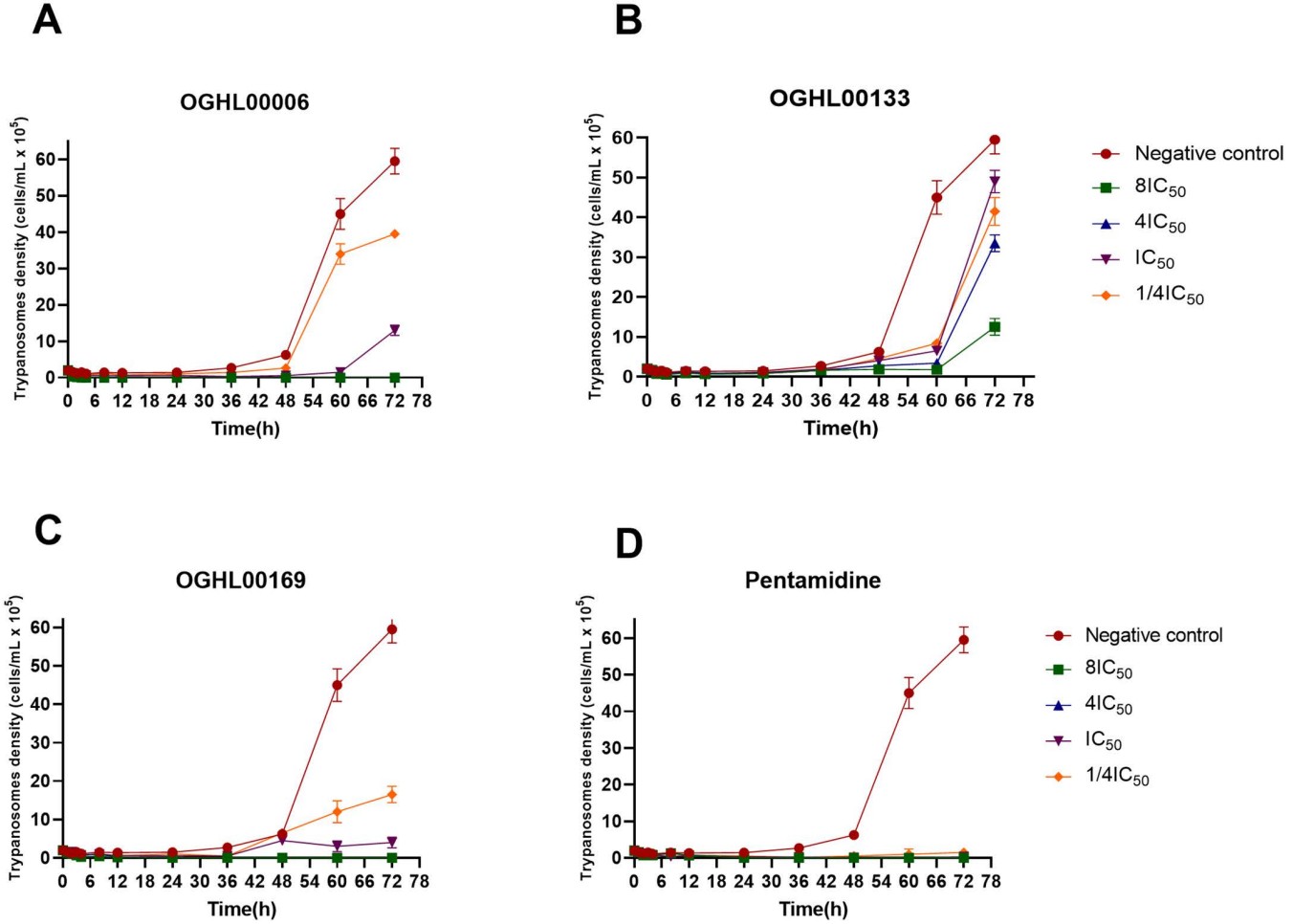

**Fig 4. Concentration-dependent killing of *T. b. b.* by OGHL00006 (A), OGHL00133 (B), OGHL00169 (C) and pentamidine (D).** All data represent means ± standard error of the mean (SEM) from two independent experiments.

OGHL00133 was docked against *Tb*KFR1 with a core (scaffold) constraint and the resulting binding pose (Fig 6A) shows that OGHL00133 forms two hydrogen bonds (H-bonds) with Ala95 of the hinge binding region (which is characteristic of ATP-competitive kinase inhibitors [33] and a π–π stacking with Phe143. The stability of the docked complex was assessed via a 100 ns MD simulation. Fig 6B shows the root mean square deviation (RMSD) evolution of the protein (left Y-axis) during the course of the simulation, with changes of the order of 0–4 Å, indicating slight conformational changes. The ligand RMSD (right Y-axis) with changes of the order of 0–1 Å indicates the ligand is stable within the binding pocket. Moreover, there is a high H-bond persistence between the 2,4-diaminopyrimidine moiety of the ligand and residue Ala95 of the hinge binding region, as well as a π–π stacking (hydrophobic interactions) with Phe143 (Fig 6C).

OGHL00006 {(E)-4-(4-((2,4-dichlorobenzyl)thio)phenyl)-N,N-dimethyl-4-oxobut-2-enamide} inhibits the human CX3C motif chemokine receptor 1 (CX3CR1; $IC_{50} = 526$ nM), protein kinase B (PKB) and the signal transducer and activator of transcription 5 (STAT5). It possesses a carbonylacrylamide moiety (Michael acceptor), known to selectively and covalently binds to reactive cysteines in protein binding sites [34]. Several studies have identified *T. brucei* kinases that possess reactive cysteines. Notably, Nishino et al. [35] performed affinity-based protein profiling to deconvolute the primary

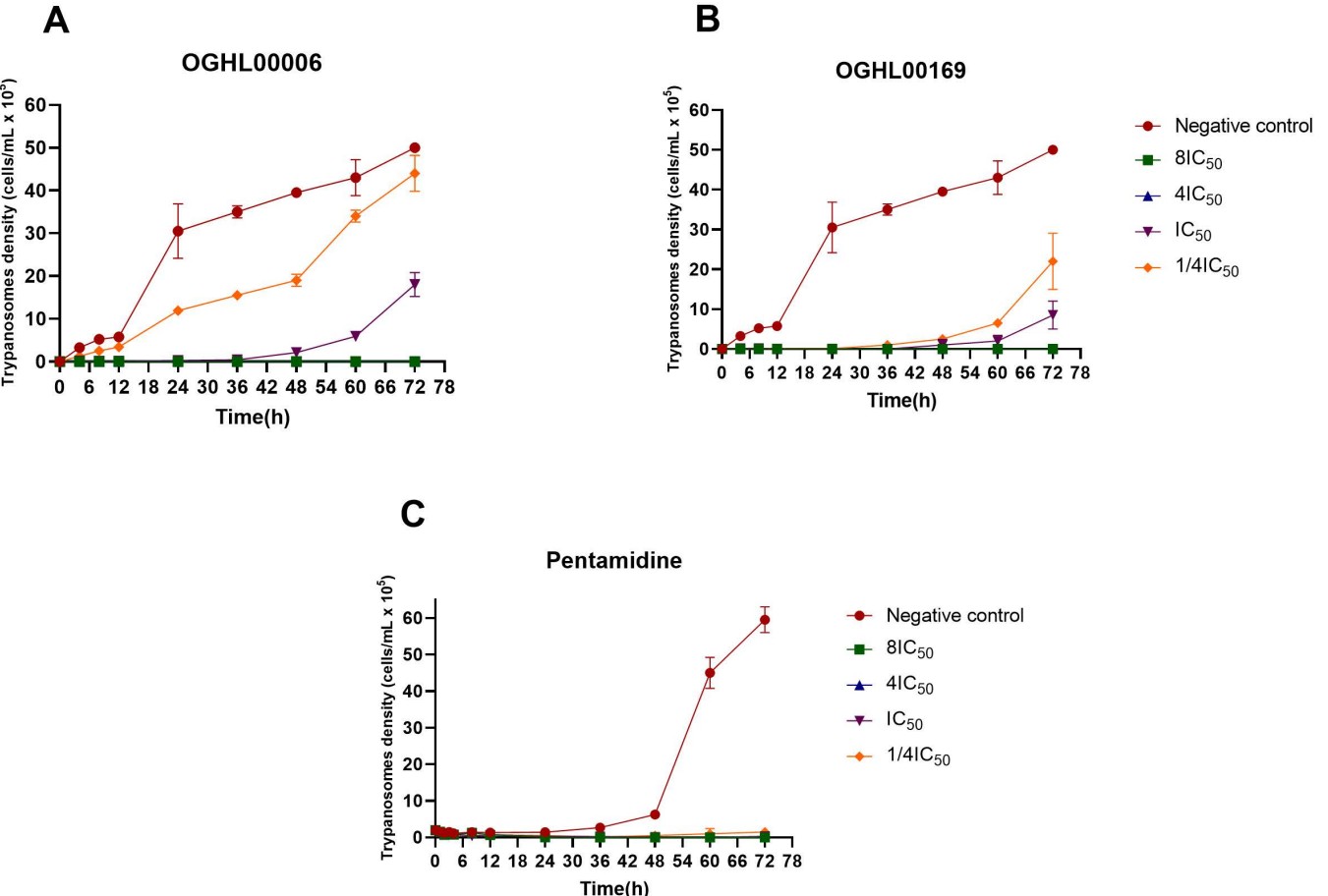

**Fig 5. *T. b. b.* survival over time following OGHL00006 (A), OGHL00169 (B) and pentamidine (C) incubations and washout. The plotted data are the means and SD of two independent replicates.**

molecular target of a Michael acceptor fungal polyketide known as hypothemycin. This study revealed that hypothemycin covalently inhibit *Tb*CLK1 (Tb927.11.12410), a kinase whose conditional knockdown in bloodstream forms results in the death of parasitic cells [32]. Moreover, Saldivia et al. [36] reported the first and only cocrystal structure of *Tb*CLK1 in complex with a highly potent Michael acceptor called AB1 (PDB ID: 6Q2A). Therefore, covalent docking of OGHL00006 against *Tb*CLK1 was performed and the resulting docking pose shows it forms a covalent bond with Cys215, a π–π stacking with Phe140, and a halogen bond with Tyr212 (Fig 7). Since carbonylacrylamides form irreversible covalent complexes with proteins, it may not be necessary to perform MD simulations to assess the stability of this complex.

OGHL00169 {3-(1-(2,3-difluorobenzyl)-1H-1,2,3-triazol-4-yl)-5-(1-methyl-1H-pyrazol-4-yl)-1H-pyrrolo[2,3-b]pyridine} is a kinase inhibitor targeting 3-phosphoinositide-dependent kinase-1 (PDK1; $IC_{50}$ 0.5 nM–1 µM) and TANK-binding kinase 1 (TBK1; $IC_{50}$ = 0.196 nM), without causing toxicity to normal cells ($CC_{50}$ 1–10 µM) [37]}. Although there exist a homolog *Tb*PDK1 (Tb927.9.4910), however its conditional knockdown in bloodstream forms by RNAi leads to slow parasitic growth [32]. Perhaps other *T. brucei* AGC kinases might come into question, especially notably *Tb*AEK1 (Tb927.3.2440) and *Tb*PK50 (Tb927.10.4940) are essential for the proliferation of *T. brucei* bloodstream forms proliferation *in vitro* [32]. These three proteins share a highly conserved kinase domain with key regulatory elements such as phosphorylation sites and hinge-binding region, among others [27]. Hence, a homology model of *Tb*PDK1 was built based on the cocrystal structure

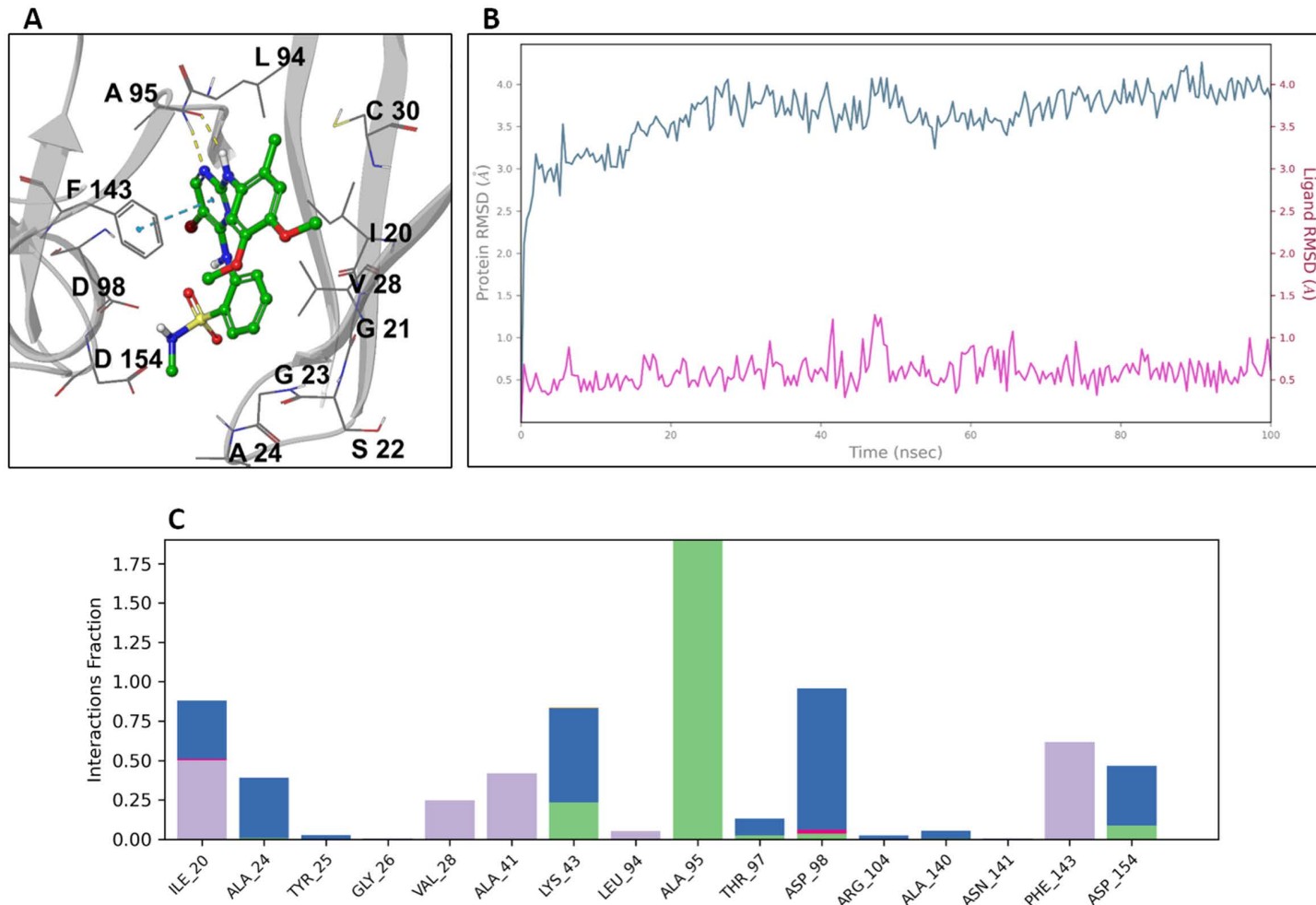

**Fig 6. Modeling of OGHL00133 against *Tb*KFR1. (A)** Docking model of the *Tb*KFR1–OGHL00133 complex; H-bonds and π–π stacking are represented as yellow and blue dotted lines, respectively. **(B)** RSMD plot of both protein and ligand during the course of a 100 ns MD simulation. **(C)** Protein-ligand interaction histogram; H-bonds, hydrophobic, ionic and water-mediated H-bonds are colored green, purple, magenta, and blue, respectively.

(PDB ID: 3RCJ) of *h*PDK1 in complex with a ligand sharing a 3-([3-(1-benzyl-1H-1,2,3-triazol-4-yl)-7-azaindole scaffold with OGHL00169. Next, OGHL00169 was core-constraint docked against *Tb*PDK1 and the resulting binding pose (Fig 8A) depicts bidendate H-bonds between the ligand and the hinge binding region (Glu96 and Cys98). The RMSD evolution of the protein (left Y-axis) during the course of a 100 ns long MD simulation (Fig 8B), with changes of the order of 0–4 Å, indicating slight conformational changes. The ligand RMSD (right Y-axis) with changes of the order of 0–2.5 Å indicates the ligand is stable within the binding pocket. Moreover, there is a high H-bond persistence between the 7-azaindole moiety of the ligand and the kinase hinge binding region (Glu96 and Cys98), as well as prominent hydrophobic contacts with Leu17, Val25, Met164, and Ile174 (Fig 8A and 8C).

## Discussion

Phenotypic screening, which involves testing compounds directly on whole cells, is a powerful approach to antitry-panosomal drug discovery. This method circumvents the need for prior knowledge of molecular targets and has been

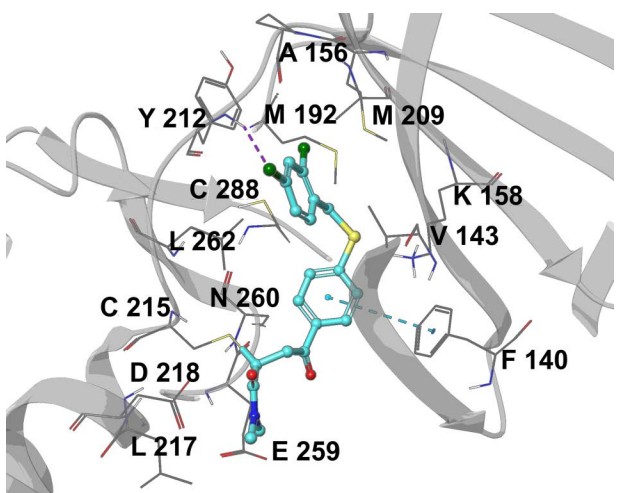

**Fig 7. Docking model of the *Tb*CLK1–OGHL00006 complex; π–π stacking and halogen bond are represented as blue, and purple dotted lines, respectively.**

pivotal in the development of clinical-stage compounds such as fexinidazole and acoziborole [38,39]. Similarly, drug repurposing, which leverages inhibitors originally developed for other disease conditions, has emerged as an efficient strategy for expanding the therapeutic arsenal against parasitic diseases. The success of this strategy is highlighted by the successful repurposing of kinase inhibitors and other enzyme inhibitors for antiparasitic research, demonstrating the overlap in molecular pathways between human diseases and parasitic infections [40]. In this study, we sought to identify antitrypanosomal small molecules for further AI-facilitated medicinal chemistry optimization and mechanisms of action investigation. Thus, screening the Merck KGaA Darmstadt Germany OGHL led to the identification of 29 inhibitors (Fig 1 and Table 1), representing several chemical classes with known antiparasitic potential (Fig 3). Among these, the 1,2,3,4-tetrahydroquinoline scaffold has demonstrated antiparasitic potential including efficacy against *Trypanosoma* species [41]. Thiazolidinone derivatives also demonstrated significant antitrypanosomal activity, with some analogues exhibiting $IC_{50}$ values of 0.6 µM when conjugated with pyrazoline moieties [42]. Same observations were made with conjugates of thiazolidinone cores [43], pyrimido[5,4-d]pyrimidine-based compounds [44], and aminopyrimidine derivatives [45]. Furthermore, [46] reported promising activity from imidazopyridine, pyrimidine-, and furopyridine-based compounds.

This mini-library has previously been utilized to identify antimicrobial agents against SARS-CoV-2 [47], *Cryptosporidium parvum* [48] and *Plasmodium falciparum* [49]. Notably, six out of the nine compounds previously reported for their antiplasmodial potency were also identified here as antitrypanosomal hits. These include OGHL00133, OGHL00134, OGHL00169, OGHL00121, OGHL00250, and OGHL00236 [49]. Recently, OGHL00006, OGHL00022, OGHL00121, and OGHL00169 emerged as the most active hits against *Schistosoma mansoni* [50]. In our current screens, OGHL00006, OGHL00133, and OGHL00169 demonstrated the most potent and selective activity against *T. b. brucei* with OGHL00006 and OGHL00169 emerging as the most promising trypanocidal hits over time (Tables 1, 2, Figs 4 and 5). They belong to a series of potent anticancer compounds. An analog of OGHL00169, a 7-azaindole was previously identify as a potent inhibitor of anaplastic lymphoma kinase (ALK) ($IC_{50}$: 90–141 nM) and aurora A kinase (84% inhibition at 100 nM), both of which are critical oncogenic drivers [51]. This aligns with the established conservation of ATP-binding sites within kinase catalytic domains [52]. Such cross-reactivity may explain why kinase inhibitors developed for cancer therapy often display antiparasitic activity. Indeed, kinases regulate critical cellular processes, and their conservation across species allows inhibitors

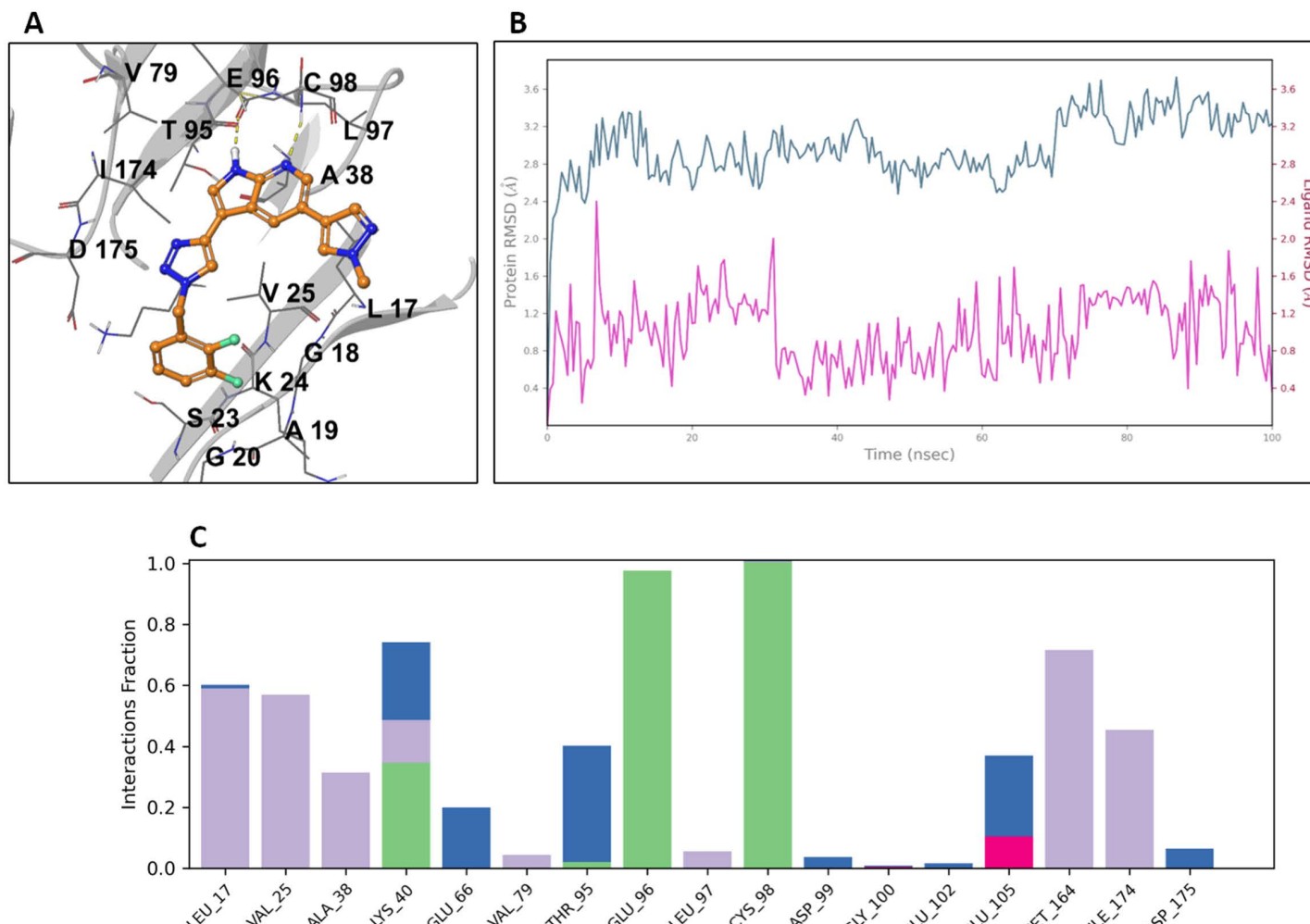

**Fig 8. Modeling of OGHL00169 against *Tb*PDK1. (A)** Docking model of the *Tb*PDK1–OGHL00169 complex; H-bonds and π–π stacking are represented as yellow and blue dotted lines, respectively. **(B)** RSMD plot of both protein and ligand during the course of a 100 ns MD simulation. **(C)** Protein-ligand interaction histogram; H-bonds, hydrophobic, ionic and water-mediated H-bonds are colored green, purple, magenta, and blue, respectively.

to disrupt parasite-specific pathways while sparing mammalian cells. For example, kinase inhibitors imatinib and dasatinib have shown efficacy against *Leishmania* and *Plasmodium* species [53,54].

The antiparasitic activity of these compounds could be rationalized based on their known targets. The metabolic similarities between cancer cells and trypanosomes, including their rapid proliferation further support the potential of anticancer drugs as broad-spectrum antitrypanosomal agents [55]. A combination of molecular modeling techniques, namely homology modeling, molecular docking, and MD simulations were employed to decipher putative mechanisms of action of the identified hits (Figs 6–8). In accordance with the observed *T. b. brucei* growth inhibition phenotypes, the primary target of OGHL00006 was proposed to be *Tb*CLK1 while that of OGHL00133 to be *Tb*KFR1. Although *Tb*PDK1 has been identified as a potential target of OGHL00169, it is unlikely to be its primary target, as RNAi-mediated conditional knockdown of this protein leads to growth arrest rather than cell death. Thus, we hypothesize that other members of the AGC kinase family, specifically *Tb*AEK1 and *Tb*PK50 — whose RNAi conditional knockdowns are associated with a cell death phenotype — may be more relevant in this context.

DMPK data of these hits further inform their development potential (Fig 9). OGHL00133 exhibited strong potency and selectivity but showed a rapid clearance rate, suggesting the need for structural modifications to enhance metabolic stability. OGHL00169 displayed promising permeability and efflux properties, making it a strong candidate for oral bioavailability. Conversely, OGHL00006 had lower solubility, which may limit its *in vivo* utility but provide a framework for chemical modifications to improve physicochemical properties. These findings validate the utility of phenotypic screening for uncovering diverse chemical scaffolds.

However, further research is warranted to explore the most promising antitrypanosomal hit compounds as scaffolds for the development of new therapies against trypanosomiasis. In particular, the *in silico* predictions require *in vitro* experimental validation on *Trypanosoma brucei* targets. Although *T. brucei brucei* shares approximately 99% genomic similarity with the two African subspecies responsible for HAT (*T. brucei rhodesience* and *T. brucei gambiense*) [56], it remains essential to confirm the efficacy of the identified hits against these human-infective strains. Furthermore, in-depth mechanistic studies are necessary to elucidate the precise mechanism of action of these compounds on *Trypanosoma brucei* pathways.

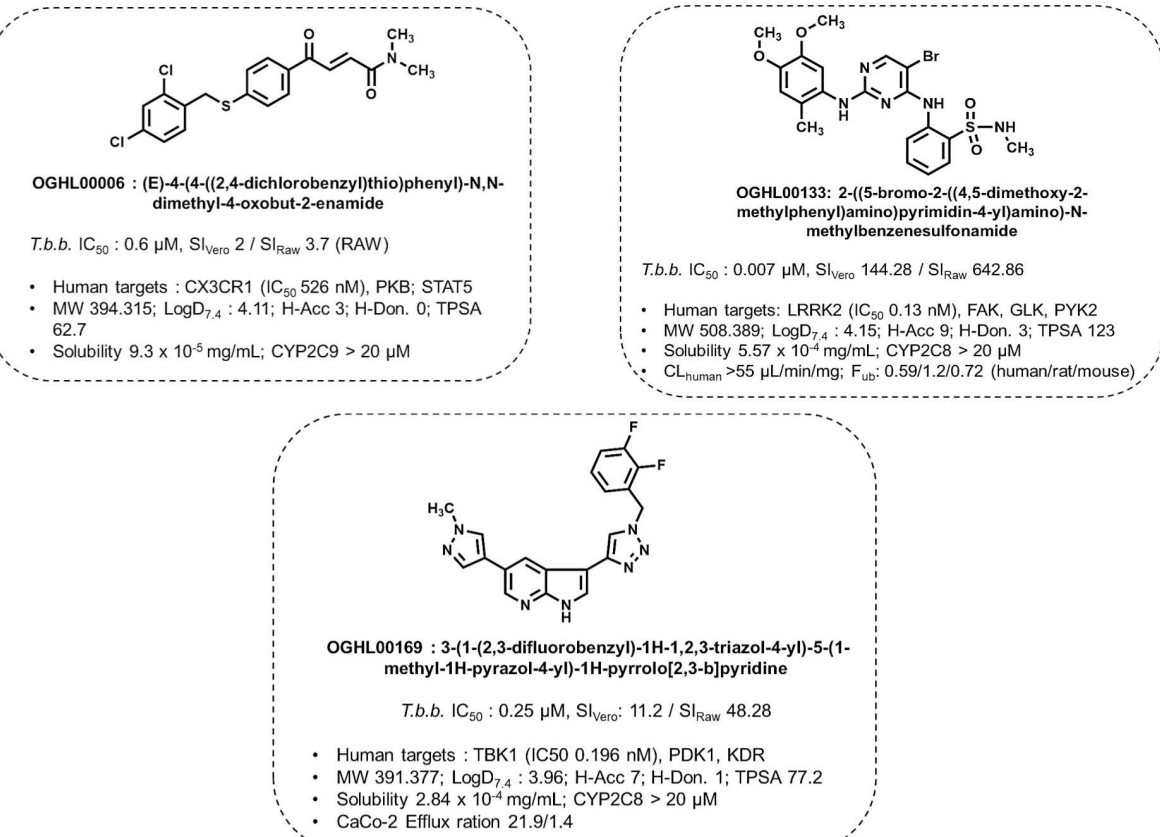

**OGHL00006 : (E)-4-(4-((2,4-dichlorobenzyl)thio)phenyl)-N,N-dimethyl-4-oxobut-2-enamide**

*T.b.b.* IC$_{50}$ : 0.6 μM, SI$_{Vero}$ 2 / SI$_{Raw}$ 3.7 (RAW)

- Human targets : CX3CR1 (IC$_{50}$ 526 nM), PKB; STAT5
- MW 394.315; LogD$_{7.4}$ : 4.11; H-Acc 3; H-Don. 0; TPSA 62.7
- Solubility 9.3 x 10$^{-5}$ mg/mL; CYP2C9 > 20 μM

**OGHL00133 : 2-((5-bromo-2-((4,5-dimethoxy-2-methylphenyl)amino)pyrimidin-4-yl)amino)-N-methylbenzenesulfonamide**

*T.b.b.* IC$_{50}$ : 0.007 μM, SI$_{Vero}$ 144.28 / SI$_{Raw}$ 642.86

- Human targets: LRRK2 (IC$_{50}$ 0.13 nM), FAK, GLK, PYK2
- MW 508.389; LogD$_{7.4}$ : 4.15; H-Acc 9; H-Don. 3; TPSA 123
- Solubility 5.57 x 10$^{-4}$ mg/mL; CYP2C8 > 20 μM
- CL$_{human}$ >55 μL/min/mg; F$_{ub}$: 0.59/1.2/0.72 (human/rat/mouse)

**OGHL00169 : 3-(1-(2,3-difluorobenzyl)-1H-1,2,3-triazol-4-yl)-5-(1-methyl-1H-pyrazol-4-yl)-1H-pyrrolo[2,3-b]pyridine**

*T.b.b.* IC$_{50}$ : 0.25 μM, SI$_{Vero}$: 11.2 / SI$_{Raw}$ 48.28

- Human targets : TBK1 (IC50 0.196 nM), PDK1, KDR
- MW 391.377; LogD$_{7.4}$ : 3.96; H-Acc 7; H-Don. 1; TPSA 77.2
- Solubility 2.84 x 10$^{-4}$ mg/mL; CYP2C8 > 20 μM
- CaCo-2 Efflux ration 21.9/1.4

**Fig 9. DMPK profile of the prioritized inhibitors.** Topological polar surface area (TPSA), hydrogen-bond donor (H-Don) and acceptor (H-Acc), Molecular weight (MW), Partition coefficient (LogD$_{7.4}$), Microsomal stability on human, rat or mice CYP450, Clearance (CL), Fraction unbound (Fu), *In vitro* permeability (CaCo-2).

## Conclusion

In conclusion, the combination of phenotypic screening and drug repurposing has identified three promising compounds with diverse rates of action and favorable selectivity against *T. b. brucei*. The *in silico* prediction data of these compounds against conserved targets between parasitic and human cells support the hypothesis that kinase inhibitors may hold potential as antiparasitic agents, although further experimental validation is required. These identified hits demonstrated typical thresholds for lead optimization, highlighting their potential as structural scaffolds for further refinement. Future studies should prioritize the validation of the proposed mechanism of action of the identified hits, prior to lead optimization.

## Supporting information

**S1 Fig. Assay plate mapping.** The mapping shows the test compounds panel, positive control (reference drug- pentamidine) panel, negative control (untreated parasites) panel, and vehicle (0.1% DMSO) panel.
(TIF)

## Acknowledgments

We thank Merck KGaA, Darmstadt, Germany, and the Open Innovation Portal for granting access to the well-plated compound library, providing solid materials for confirmatory studies, and sharing valuable data on the standout compounds.

## Author contributions

**Conceptualization:** Fabrice Fekam Boyom.

**Data curation:** Darline Dize, Aurélien F.A. Moumbock, Vianey C. Tchuenguia, Germaine Y. Bougnogolo, Fride S.B. Nana, Sandra D. W. Monkam.

**Formal analysis:** Darline Dize, Aurélien F.A. Moumbock, Sandra D. W. Monkam.

**Funding acquisition:** Fabrice Fekam Boyom.

**Investigation:** Darline Dize, Aurélien F.A. Moumbock, Vianey C. Tchuenguia, Germaine Y. Bougnogolo, Fride S.B. Nana, Sandra D. W. Monkam.

**Methodology:** Darline Dize, Aurélien F.A. Moumbock, Vianey C. Tchuenguia, Germaine Y. Bougnogolo, Fride S.B. Nana, Sandra D. W. Monkam.

**Project administration:** Fabrice Fekam Boyom.

**Resources:** Fabrice Fekam Boyom.

**Software:** Darline Dize, Aurélien F.A. Moumbock.

**Supervision:** Fabrice Fekam Boyom.

**Validation:** Darline Dize.

**Visualization:** Darline Dize.

**Writing – original draft:** Vianey C. Tchuenguia, Germaine Y. Bougnogolo, Fride S.B. Nana, Sandra D. W. Monkam.

**Writing – review & editing:** Darline Dize, Aurélien F.A. Moumbock, Fabrice Fekam Boyom.

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
