## [Decision Letter · Decision Letter 0]

8 Sep 2025

Potent Kinase Inhibitors from the Merck KGaA OGHL: Novel Hits against Trypanosoma brucei with Potential for Repurposing

Dear Dr. Boyom,

Thank you for submitting your manuscript to PLOS Neglected Tropical Diseases. After careful consideration, we feel that it has merit but does not fully meet PLOS Neglected Tropical Diseases's publication criteria as it currently stands. Therefore, we invite you to submit a revised version of the manuscript that addresses the points raised during the review process.

Please submit your revised manuscript within 60 days Nov 07 2025 11:59PM. If you will need more time than this to complete your revisions, please reply to this message or contact the journal office at plosntds@plos.org. Please include the following items when submitting your revised manuscript:

We look forward to receiving your revised manuscript.

Kind regards,

Sarman Singh, MD, FRSC, FRCP

Section Editor

Sarman Singh

Section Editor

Shaden Kamhawi

co-Editor-in-Chief

Paul Brindley

co-Editor-in-Chief

**Additional Editor Comments:**

I tend to agree with comments of the reviewer that the study needs clearer methodological details. Specifically, (1) please include a vehicel control to satisfy the author concerns. (2) it will be better to add one plate layout ( as supplementary material) of the drugs and control included in the plate and (3) make the acknowledgement more explicit to Merck KGaA, Darmstadt, Germany, and the Open Innovation Portal, preferably in methodology section provide their portal link and how they help the research of new molecules.

**Journal Requirements:**

Please upload all main figures as separate Figure files in .tif or .eps format. For more information about how to convert and format your figure files please see our guidelines: 

**Reviewers' Comments:**

Reviewer's Responses to Questions

**Key Review Criteria Required for Acceptance?**

**Methods:**

-Are the objectives of the study clearly articulated with a clear testable hypothesis stated?

-Is the study design appropriate to address the stated objectives?

-Is the population clearly described and appropriate for the hypothesis being tested?

-Is the sample size sufficient to ensure adequate power to address the hypothesis being tested?

-Were correct statistical analysis used to support conclusions?

-Are there concerns about ethical or regulatory requirements being met?

Reviewer #1: The methodology section is unclear. It missed important details.

Line 115: DMSO has cytotoxic effect on the cells. Justify its use as a solvent in your work on cell cultures. Moreover, it may have an anti-parasitic effect on Trypanosomes. Mention how to exclude this effect in your study.

Line 118: Define T. b. brucei Lister 427 as a virulent lab strain of Trypanosoma brucei

Line 120: add the principle of resazurin reduction assay

What are the reference drugs and the negative control?

Lines 138 and 139: for a full dose-response assay, a single concentration of each compound and the positive control is not sufficient . This is not a proper control for a dose-response curve, as a dose-response curve requires multiple concentrations.

Line 139: The incubation time for the cytotoxicity assay is stated as 42 hours, which is an unusual duration. Most cell viability assays use either 24, 48, or 72 hours. This should be justified.

Line 148: state the actual concentrations used

Line 154: Missing reagent in the sentence ”washed three times with…”

Line 155: "1/100 drug dilution," is contradictory to resuspending in "drug-free medium.

Line 150 and 156: re write the enumeration of motile cells

Line 227: reevaluation of the powdered forms of some compounds should be added in the methodology.

Why did the authors choose these three compounds only to test for the powdered forms?

Line 228: It is not clear which reagent was used to dissolve them.

Reviewer #2: The introduction clearly demonstrates the urgent need for new trypanocidal agents and the study objective is to identify novel hits from the Merck KGaA Open Global Health Library (OGHL) and rationalize their mechanisms

Hypothesis is Implicit but testable.

Study design aligns with the stated objectives, SI values were calculated, and data were presented with mean ± SD.

Since the work used in vitro parasite and mammalian cell cultures, there are no direct human/animal subjects requiring ethics approval.

**Results:**

-Does the analysis presented match the analysis plan?

-Are the results clearly and completely presented?

-Are the figures (Tables, Images) of sufficient quality for clarity?

Reviewer #1: The results section is redundant and confusing

It lacks documentation of different assays and cultures used.

What are the effect of the tested compounds on trypanosomes themselves?

Reviewer #2: The analysis plan described follows the Results and Discussion

Yes. Results are broken into clear sections (screening hits, SAR, time-kill, molecular docking, DMPK).

Tables are detailed with IC₅₀/CC₅₀ values, figures (dose-response, SAR structures, docking models) are of publishable quality

**Conclusions:**

-Are the conclusions supported by the data presented?

-Are the limitations of analysis clearly described?

-Do the authors discuss how these data can be helpful to advance our understanding of the topic under study?

-Is public health relevance addressed?

Reviewer #1: Some conclusions are not supported by the data presented. for example, Line 546: The authors mentioned that “The ability of these compounds to disrupt conserved pathways” . This conclusion is based on which results???

What are the limitations of the current study?

Reviewer #2: This research paper is highly relevant to public health. The introduction and author summary emphasize that African trypanosomiasis remains a critical health and socio-economic burden, complicated by drug resistance and limited treatment options.

Some limitations are acknowledged, but they need clearer articulation.

I recommend adding a short, explicit “Limitations” paragraph in the Discussion.

**Editorial and Data Presentation Modifications?**

Reviewer #1: (No Response)

Reviewer #2: Accept after minor revision

**Summary and General Comments:**

Reviewer #1: The authors presented a manuscript regarding the effect of different compounds. However, a detailed methodology and clear results should be presented.

Reviewer #2: Overall, this manuscript does address all the checklist points. The only possible gap is the lack of an explicitly worded hypothesis statement and formal statistical power justification. Everything else, objectives, design, population, methods, analyses, and results is clear and appropriate.

PLOS authors have the option to publish the peer review history of their article (what does this mean? ). If published, this will include your full peer review and any attached files.

**Do you want your identity to be public for this peer review?** For information about this choice, including consent withdrawal, please see our Privacy Policy .

Reviewer #1: No

Reviewer #2: **Yes: ** Ozioma Onuselogu

**Figure resubmission:**
---

## [Decision Letter · Decision Letter 1]

22 Oct 2025

Response to Reviewers
Revised Manuscript with Track Changes
Manuscript

Shaden Kamhawi

co-Editor-in-Chief

Paul Brindley

co-Editor-in-Chief

**Reviewers' comments:**

**Key Review Criteria Required for Acceptance?**

**Methods**

-Are the objectives of the study clearly articulated with a clear testable hypothesis stated?

-Is the study design appropriate to address the stated objectives?

-Is the population clearly described and appropriate for the hypothesis being tested?

-Is the sample size sufficient to ensure adequate power to address the hypothesis being tested?

-Were correct statistical analysis used to support conclusions?

-Are there concerns about ethical or regulatory requirements being met?

Reviewer #1: Line 162: Use the sentence provided in your response [The three compounds were selected for confirmatory studies based on their highest antitrypanosomal activity (IC50 ≤1μM) and good selectivity (SI ≥50 μM)]

instead of

[Confirmatory dose-response assays were conducted for the most active and selective compounds identified during the primary screening].

Reviewer #2: (No Response)

**Results**

-Does the analysis presented match the analysis plan?

-Are the results clearly and completely presented?

-Are the figures (Tables, Images) of sufficient quality for clarity?

Reviewer #1: (No Response)

Reviewer #2: (No Response)

**Conclusions**

-Are the conclusions supported by the data presented?

-Are the limitations of analysis clearly described?

-Do the authors discuss how these data can be helpful to advance our understanding of the topic under study?

-Is public health relevance addressed?

Reviewer #1: (No Response)

Reviewer #2: (No Response)

**Editorial and Data Presentation Modifications?**

Reviewer #1: (No Response)

Reviewer #2: (No Response)

**Summary and General Comments**

Reviewer #1: (No Response)

Reviewer #2: Overall, looks good!

PLOS authors have the option to publish the peer review history of their article (what does this mean? ). If published, this will include your full peer review and any attached files.

**Do you want your identity to be public for this peer review?** For information about this choice, including consent withdrawal, please see our Privacy Policy .

Reviewer #1: **Yes: ** Dalia S. Ashour

Reviewer #2: **Yes: ** Ozioma Esther Onuselogu

**Figure resubmission:**

**Reproducibility:** To enhance the reproducibility of your results, we recommend that authors of applicable studies deposit laboratory protocols in protocols.io, where a protocol can be assigned its own identifier (DOI) such that it can be cited independently in the future. Additionally, PLOS ONE offers an option to publish peer-reviewed clinical study protocols. Read more information on sharing protocols at https://plos.org/protocols?utm_medium=editorial-email&utm_source=authorletters&utm_campaign=protocols

---

## [Editor Report · Decision Letter 2]

4 Nov 2025

Dear Dr. Boyom,

We are pleased to inform you that your manuscript 'Potent Kinase Inhibitors from the Merck KGaA OGHL: Novel Hits against Trypanosoma brucei with Potential for Repurposing' has been provisionally accepted for publication in PLOS Neglected Tropical Diseases.

Best regards,

Sarman Singh, MD, FRSC, FRCP

Section Editor

Sarman Singh

Section Editor

Shaden Kamhawi

co-Editor-in-Chief

Paul Brindley

co-Editor-in-Chief

Congratulations.

---

## [Editor Report · Acceptance letter]

Dear Dr. Boyom,

We are delighted to inform you that your manuscript, " 

Potent Kinase Inhibitors from the Merck KGaA OGHL: Novel Hits against Trypanosoma brucei with Potential for Repurposing," has been formally accepted for publication in PLOS Neglected Tropical Diseases.

Best regards,

Shaden Kamhawi

co-Editor-in-Chief

Paul Brindley

co-Editor-in-Chief
